# MULTI-SCALE NETWORK ARCHITECTURE SEARCH FOR OBJECT DETECTION

## ABSTRACT

Many commonly-used detection frameworks aim to handle the multi-scale object detection problem. The input image is always encoded to multi-scale features and objects grouped by scale range are assigned to the corresponding features. However, the design of multi-scale feature production is quite hand-crafted or partially automatic. In this paper, we show that more possible architectures of encoder network and different strategies of feature utilization can lead to superior performance. Specifically, we propose an efficient and effective multi-scale network architecture search method (MSNAS) to improve multi-scale object detection by jointly optimizing network stride search of the encoder and appropriate feature selection for detection heads. We demonstrate the effectiveness of the method on COCO dataset and obtain a remarkable performance gain with respect to the original Feature Pyramid Networks.

## 1 INTRODUCTION

Recognizing and localizing objects at vastly different scales is a fundamental challenge in object detection. Detection performance for objects with different scales is highly related to features with different properties such as feature resolution, receptive fields, and feature fusion ways. The key to solving the multi-scale problem in object detection is how to build a multi-scale network that has proper high-level semantic features for objects with different scales.

A recent work in object detection Feature Pyramid Networks(FPN) (Lin et al., 2017) has achieved remarkable success in multi-scale feature design and has been commonly used by many modern object detectors (He et al., 2017; Lin et al., 2020; Lu et al., 2019). FPN extracts multi-scale intermediate features from the encoder network and assigns objects grouped by scales to corresponding features according to a heuristic rule. Another prevalent detection framework, SSD (Liu et al., 2016), conducts feature generation by a lighter encoder network without upsampling operators. The basic idea to deal with the multi-scale detection problem can be summarized as below. Given the input image, a series of feature maps with the various resolution are generated to detect objects grouped by scale range. We note it as multi-scale feature production. In FPN and its variants, the multi-scale feature production is split into two steps, feature generation and feature utilization. In terms of feature generation, an encoder network composed of blocks provides features with different scales. And the strategy of feature utilization determines the rule of assigning objects to feature maps. These two steps are closely related to each other.

Although FPN has achieved promising results on multi-scale object detection tasks, the production of multi-scale features is quite hand-crafted and relies heavily on the experiences of human experts. More specifically, network architectures of FPN are based on a downsample-upsample architecture which may not be effective enough. By changing the downsampling and upsampling operation's positions and numbers, we could obtain many other candidates to generate different multi-scale features. Also, the predefined rule of feature utilization is very empirical and other alternatives may lead to better performance. Therefore we wonder: Can we find network architectures that can build better semantic feature representation for multiple scales? The answer is yes.

Recent advances in neural architecture search have shown promising results compared with hand-crafted architecture by human experts (Zoph et al., 2018; Liu et al., 2019b; Cai et al., 2019; Guo et al., 2019). Several works have also focused on neural architecture search in object detection tasks (Chen et al., 2019; Ghiasi et al., 2019; Du et al., 2019), but generating and utilizing multi-scale

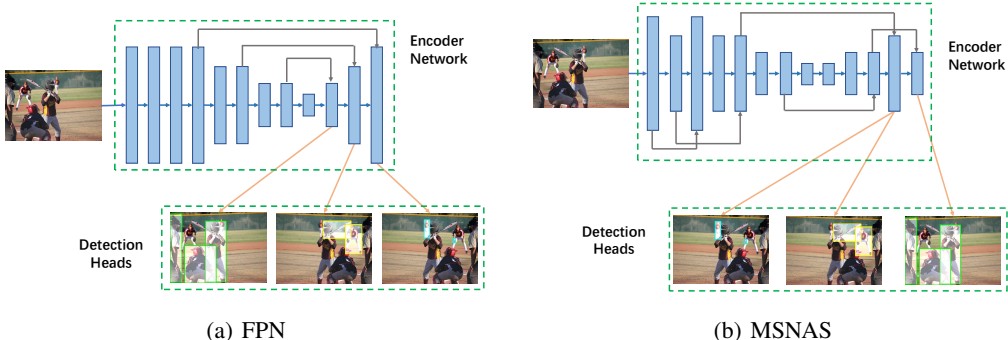

(a) FPN                                        (b) MSNAS

Figure 1: Architecture of ResNet18-FPN and the searched network of MSNAS-R18. MSNAS-R18 has different stride values of blocks in the encoding network and a more flexible feature utilization strategy. For simplification, P6 of FPN is not included in the figure and only three detection heads are presented.

features are still not well explored. DetNAS (Chen et al., 2019) adopts the method mainly designed on image classification to search the operations of backbone networks in object detectors. NAS-FPN (Ghiasi et al., 2019) focuses on searching for better feature-fusion connections in the neck part of FPN. NAS-FPN doesn't optimize the whole encoder network and still relies on predefined backbone architecture. Recently SpineNet (Du et al., 2019) proposes a search method with scale-permuted features and cross-scale connections by reinforcement learning, but the search cost is quite large. All these previous works focus on designing better neural network architectures to generate better features given a fixed feature selection strategy. However, they fail to conduct a complete flexible multi-scale feature production strategy.

In this paper, we propose a new method to take into account of both aspects and build detection networks with the strong and proper multi-scale feature production strategy by neural architecture search. For feature generation, we put forward a network stride search method to generate multiple feature representations for different scales. Different from the scale-decreasing-increasing architecture of FPN, the scale of our networks can decrease or increase at each block, as illustrated in Figure 1. By stride search for each block, we could significantly explore a wide range of possible feature generation designs of multi-resolution networks. Most backbones of object detectors are originally designed on image classification without multi-scale problems. However, stride configuration in the encoder network would be optimized in the context of the multi-scale task. Moreover, more complex cross-scale feature fusions might appear according to more complex internal scale changes. For feature utilization, we change the previous one-to-one mapping strategy into a more flexible feature selection. Since each group with objects of the same scale range owns one detection head, feature utilization is implemented by selecting proper features for detection heads. Objects of different scale ranges might be assigned to the same feature map. It is not possible in previous methods, as shown in Figure 1(b).

By jointly optimizing feature generation and utilization of multi-scale features, we search for flexible but complete multi-scale feature production strategies. Extensive experiments demonstrate complete multi-scale feature production search is critical to building strong and proper semantic features for object detection with different scales. On challenging COCO dataset (Lin et al., 2014), our method obtains a 2.6%, 1.5%, 1.2% mAP improvement with similar FLOPs as ResNet18-FPN, ResNet34-FPN, ResNet50-FPN.

## 2  RELATED WORK

**Neural Architecture Search** Neural Architecture Search aims to design better network architectures automatically. RL-based methods (Zoph et al., 2018; Zoph & Le, 2017) have achieved great success despite a huge computation cost. In differentiable algorithms (Liu et al., 2019b; Cai et al., 2019), architecture parameters are employed and operators in the search space are considered as

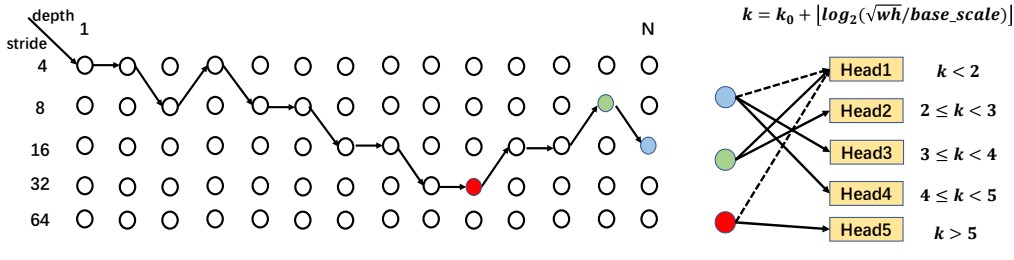

(a) Super-net and one of the paths of feature generation

(b) One example of feature utilization

Figure 2: Figure (a) shows the basic architecture of stride selection in the super-net for feature generation. The path represents one of its sub-architectures. The colored nodes are corresponding output features of the encoder network. Figure (b) illustrates the feature utilization search. The formula in Figure (b) represents the rule to assign RoIs to multi-scale features in FPN. We use different ranges of $k$ to distinguish different object groups as well as their detection heads. The solid lines show an example of feature utilization strategies. The dotted lines imply each detection head can select any of the output features.

the weighted sum of candidate operators. There exist difficulties to deal with operators with different strides. Super-nets, acting as the collection of weights shared by all the sub-architectures, and evolutionary search are involved in one-shot NAS (Guo et al., 2019; Bender et al., 2018). But it's difficult to ensure strong correlations between one-shot and stand-alone performances of the sub-architectures.

**Multi-scale Object Detection** SSD (Liu et al., 2016) uses multi-scale features generated by different stages of the backbone network to detect objects of different scales. Feature pyramid architectures are utilized in FPN (Lin et al., 2017) and RetinaNet (Lin et al., 2020) to obtain multi-scale features. SNIP (Singh & Davis, 2018) includes the image pyramid architecture to deal with multi-scale detection. Frameworks with multi-scale features are prevalent as objects of different scales appear in one image.

**Neural Architecture Search on Object Detection** DetNAS (Chen et al., 2019), NAS-FCOS (Wang et al., 2019) and Auto-FPN (Xu et al., 2019) focus on the architecture of the top-down pathway and feature fusion. SM-NAS (Yao et al., 2019) and CR-NAS (Liang et al., 2020) try to adjust the computation occupied by different parts of detectors. Also, there are several works aiming to improve FPN using NAS. NAS-FPN (Ghiasi et al., 2019) improves detection performance by searching for better connections within the feature pyramid network. It is limited without modification to the overall encoder architecture. And it fails to take the multi-scale feature utilization into account. Efficient-Det (Tan et al., 2019) and Auto-FPN (Xu et al., 2019) search better feature fusion for FPN with differentiable methods. Other relative works like Liu et al. (2019a) conduct similar modifications. Recently, SpineNet (Du et al., 2019) proposes a backbone search method with scale-permuted features and cross-scale connections by reinforcement learning. Our work has several major differences from it. First, the search space of MSNAS is designed that each operator in the network can downsample or upsample instead of permutation, which builds a much larger search space than SpineNet. Second, the complete multi-scale feature production is considered in our work, while SpineNet only focuses on the architecture of the encoder network. Lastly, our method is based on the one-shot search method instead of reinforcement learning in SpineNet. Our method is much more efficient and requires much less computation cost than SpineNet.

## 3 METHOD

We start by discussing multi-scale feature production for the object detection network in Section 3.1. In Section 3.2 we will introduce how to build the search space and search proper stride in detection networks to obtain better multi-scale features. In Section 3.3, how to search the appropriate strategy

of feature utilization is presented. Finally, details about super-net training and search strategy are described in Section 3.4.

## 3.1 FEATURE PRODUCTION FOR DETECTION NETWORK

In this section, we'll discuss the production of multi-scale features and define the problem in detail. As noted above, the basic idea of handling the multi-scale detection problem can be summarized as feature production. In feature production, the input image is first encoded into a series of feature maps with various resolutions. Then the objects are detected based on the features according to their scales. One method is to produce the feature for each scale range by one neural network, like the featurized image pyramid discussed in Lin et al. (2017). Variants include utilizing the image pyramid, like SNIP (Singh & Davis, 2018). However, we employ only one neural network and obtain multi-scale features from intermediate features of the network. Because deep neural networks are experts in encoding the image into features. And they are considered to be able to encode information of different scales into features with different resolutions. Then we face the problem of how to utilize these features since there are $N$ features available for $K$ object groups. Therefore, it is reasonable to split feature production into feature generation and feature utilization. To be more specific, the problem of multi-scale feature production can be defined as Equation 1. When we divide the problem into feature generation and feature utilization, as in Equation 2 and Equation 3, $\phi$ is approximated by $g \circ f$.

$$\phi : \mathbb{R}^{3 \times W \times H} \to \{\mathbb{R}^{H_i \times W_i \times H_i}\}_K \tag{1}$$

$$f : \mathbb{R}^{3 \times W \times H} \to \{\mathbb{R}^{H_i \times W_i \times H_i}\}_N \tag{2}$$

$$g : \{\mathbb{R}^{H_i \times W_i \times H_i}\}_N \to \{\mathbb{R}^{H_i \times W_i \times H_i}\}_K \tag{3}$$

And it is likely that only optimizing feature generation, as previous works do, is not optimal. So instead of optimizing feature generation for all the feature utilization strategies, we jointly optimize feature generation and utilization as a whole.

## 3.2 FEATURE GENERATION

Resolution of feature maps in one network changes with downsampling or upsampling operators. The network architectures of FPN follow the downsampling-upsampling style, as Figure 1(a) shows. By encouraging a more flexible design of the scale-changing operation's positions and numbers, we could obtain many more promising candidates to generate better multi-scale features. We implement that by searching the stride of each block in the network.

**Search space** By deconstructing and generalizing the prevalent feature pyramid architecture, the basic search space is built as a stride-variable straight structure. A super-net of MSNAS with the depth of N consists of N mixed-blocks, $MB_1, MB_2, ..., MB_N$. For each mixed-block, three possible strides are provided, i.e. 0.5, 1, and 2. The block whose stride equals to 0.5 is implemented as an upsampling block with an interpolation operator followed by a convolution to double the width and height of the feature map. Blocks that don't change the resolution of the feature map are referred to as normal blocks. The resolution of the feature output by one mixed-block could be twice, half or the same as the input feature, as illustrated in Figure 2(a). Considering operators with different strides within a mixed-block and the variation of sizes output from different operators in one mixed-block, the super-net is designed as a path-wise structure like Guo et al. (2019). One path in the super-net is treated as valid if none of the blocks output feature larger than a quarter or smaller than 1/64 of the input image. Invalid paths are removed either in the training process of super-net or the sampling during the evolutionary search.

**Lateral connections** Lateral connections are built according to current sub-architecture, as Figure 1(b) shows. One additional 1×1 lateral convolution attached after every mix-block is available for latter cross-block connections. In scale-decreasing architectures, blocks can be grouped by resolution of output features. Each group is notated as one stage. Similarly, we refer stage to a group of adjacent blocks with the same output resolution, e.g. one downsampling or upsampling block and following normal blocks. The feature map of the last mix-block at one stage is merged with the lateral feature by element-wise addition as Equation 4-Equation 6 shows.

$$x_i = MB_i(x_{i-1}) + lat_i \tag{4}$$

$$lat_i = \begin{cases} LateralConv_{ri}(x_{ri}), & \text{if } (stride_{i+1} \neq 1) \text{ or } (i = N - 1) \\ 0, & \text{otherwise} \end{cases} \tag{5}$$

$$\text{where } ri = \max\left(\{k \mid (\prod_{k < j \leq i} stride_j = 1) \text{ and } (stride_{k+1} \neq 1)\}\right) \tag{6}$$

where $lat_i$ means the lateral connection of block $i$ to combine with. If block $i$ is the last block of a stage or the end of the encoder network as described in Equation 5, $lat_i$ is generated by lateral convolution $LateralConv_{ri}$. Among blocks with output feature of the same resolution as block $i$, block $ri$ is the nearest one at a different stage. It can be regarded as an extensive version of lateral connections in the original FPN structure.

## 3.3 Feature Utilization

In most multi-scale detection frameworks, objects are assigned to feature maps according to their scales given a predefined strategy. In this section, we will discuss how to build the search space of feature utilization with respect to the feature generation network. Basically, objects are split into $G$ groups and there exists one detection head for each group. So feature utilization strategy could be simplified by selecting the resolution of feature maps from generated multi-scale features for each detection head. Figure 2(b) shows one example of feature utilization. Three feature maps of different resolutions are available. In this case, objects in various scale ranges might be assigned to the same feature map. This is very different from previous predefined strategies. The dotted lines represent a possibility of connecting to features of any resolution provided by the encoder network.

When searching for feature utilization, the exploration to obtain better features of object groups is implemented within a lessened search space. For convenience, several constraints are designed for more efficient search, as Equation 7 shows.

$$\begin{aligned} \mathbf{s}_i \leq \mathbf{s}_{i+1} \; &\forall 0 \leq i < G \\ \min \mathbf{s} &\leq 8 \\ \max \mathbf{s} &\geq 8 \\ \min \mathbf{s} &\neq \max \mathbf{s} \end{aligned} \tag{7}$$

Let $\mathbf{s}$ be an array with length of $G$. $G$ equals to the number of object groups as described above. The $i$th item in $\mathbf{s}$, noted as $\mathbf{s}_i$, represents the selected size with respect to the input image of the feature for the corresponding object group. For example, $\mathbf{s} = (4, 8, 16, 32, 64)$ is the configuration counterpart of FPN. $\mathbf{s}$ is assumed as a monotonic sequential based on insights to assign multi-scale objects. That is to say, smaller objects are considered to be assigned to finer-resolution features, while larger ones are more compatible with coarser ones. Besides, the degraded pyramid structures are excluded in the super-net. We expect to focus more on hierarchical architectures and avoid extreme memory consumption of some special architectures.

## 3.4 Super-net Training and Search Strategy

It's difficult to combine features with different resolutions by element-wise addition, so one-shot based search strategies show great compatibility with our search space. During training the super-net, one valid path, fulfilling all the requirements in Section 3.2 and Section 3.3, is randomly sampled to optimize weights in the super-net. Inspired by Zhang et al. (2020), we treat the super-net as a good pre-trained model. A better rank could be obtained within a few iterations of individual fine-tuning, although the primitive weights in the super-net don't perform well in terms of ranking random samples. Fine-tuning for each architecture individually for a few iterations not only modulates the globally-optimized shared weights towards more personalized weights but also modifies the statistics of batch normalization operators. And the additional computation cost is marginal in the entire pipeline.

Evolutionary search is adopted after the super-net training as Algorithm 1 shows. The function $GetValidRandomSample(n)$ returns n valid random samples as described in the last paragraph. The evolution process starts from a population with size $P$. Variation operations are performed on both the encoder and stride of heads' selected features. In Algorithm 1, $CrossoverEncoder$ means doing crossover concerning the stride values in the encoder network and $CrossoverFeatureStride$ means doing a crossover concerning the selected stride values of utilization. $MutationEncoder$

and $MutationFeatureStride$ have similar meanings about doing mutation. Given that the set of valid children within computation constraints is not continuous with respect to crossover and mutation, not enough children could be generated in some iterations and the search process would terminate at a local optimum of the search space. Like Liu et al. (2020), a random set of new children from $R$ attempts of valid samples with various computations are appended as the proposals of population. In this way, both exploitation and exploration in the search space are encouraged to be conducted.

---

**Algorithm 1:** Evolution Process

---

**Input:** population size $P$, total evolution iteration $T$, max variation attempts $M$, attempts of random children $R$, Constraints $C$, return top samples $k$

**Output:** the top architectures with the best one-shot performances that meet both the validity requirements and computation constraints

1   $pop_0 := GetValidRandomSample(P)$;
2   **for** $i = 1 : T$ **do**
3     $pop_i := \emptyset$;
     // Generate children by crossover and mutation
4     $j := 0$;
5     **while** $j < M$ *and* $|pop_i| \leq P$ **do**
6       $children := CrossoverEncoder(pop_{i-1}) \cup MutationEncoder(pop_{i-1}) \cup$ $CrossoverFeatureStride(pop_{i-1}) \cup MutationFeatureStride(pop_{i-1})$;
       $children := Select(children, C)$;
7       $pop_i := pop_i \cup children$;
8       $j := j + 1$;
9     **end**
10     // Add random children
11     $random\_children := GetValidRandomSample(R)$;
12     $random\_children := Select(random\_children, C)$;
13     $pop_i := pop_i \cup random\_children$;
14     $pop_i := Topk(pop_i \cup pop_{i-1}, P)$;
15 **end**
16 **return** $Topk(pop_T, k)$

---

## 4 EXPERIMENTS

Experiments are presented in the following sections. Section 4.1 describes the implementation details. Section 4.2 shows the main results of MSNAS along with their FPN baselines. Ablation experiments are conducted and discussed in Section 4.3. Finally, the performance of MSNAS and comparison with other methods are included in Section 4.4.

### 4.1 IMPLEMENTATION DETAILS

**Dataset** COCO (Lin et al., 2014) is one of the commonly used dataset for object detection and instance segmentation. It contains a training set with around 118K images, a validation set with around 5K images, and a test-dev set with about 20k images. The annotations cover 80 categories of common objects.

**Super-net training details** We train our super-net and retrain the best architectures in the same settings. An input image is resized so that the shorter side is no more than 800 and the longer side is less than 1333, then both sides will be fulfilled by padding to be divided by 64. The models are trained from scratch for 4x-long time with a batch size of 32 images on 16 GPUs. The learning rate is initialized as 0.00125 and increases to 0.04 after a warm-up epoch. Then it is divided by 10 at the 42nd and 47th epoch. The weight decay is set to 1e-4 and the momentum is 0.9. Each architecture sample is fine-tuned for several iterations(100 iterations) with a batch size of 32 at a learning rate of 0.004 before testing and evaluation. The evolutionary search process is repeated for 20 iterations. The population size is 50 and 50 children are generated to update the population in each iteration. Only those with computation within a 1% gap of the target FLOPs will be considered as valid

Table 1: Experimental results with respect to their FPN baselines.

| Baseline Architectures | Baseline mAP | FLOPs (Encoder+RPN) | MSNAS (Ours) | Mean mAP | Var mAP | Max mAP |
|---|---|---|---|---|---|---|
| R18-FPN | 34.6 | 145.45 | MSNAS-R18 | 36.94 | 0.03 | 37.2 |
| R34-FPN | 37.7 | 182.20 | MSNAS-R34 | 38.86 | 0.0784 | 39.2 |
| R50-FPN | 38.3 | 197.33 | MSNAS-R50 | 39.3 | 0.028 | 39.5 |

children. As we focus on the design of the encoder network and multi-scale feature utilization, only the computations of encoder and RPN head are involved during the search. So in 1, the FLOPs of the first stage of the network are used. For better comparison, we report our results with FLOPs of the entire network, as shown in 6. Since the real image input size differs from image to image as indicated in the previous part, a fixed approximate input size is used when computing FLOPs of the architectures. In both 6 and 1, the approximate value for architectures with $800 \times 1333$ input is set to $832 \times 1280$ and that for networks with $600 \times 1000$ input is $576 \times 1024$. About 10 individuals will be appended to the population randomly in every iteration. Finally, five of the top samples after the search procedure are retrained to compute the statistics.

**Detection network details** Following He et al. (2019), the b-box head at the second stage originally composed of two fully-connected layers is replaced by a structure with two convolutions and one fully-connected layer. We adopt synchronized batch normalization to both the encoder and the b-box head. Blocks with different strides share the same number of channels, which we adjust to get a proper distribution of computation in the search space. An ideal search space includes a large proportion of architectures with similar FLOPs as the target FLOPs. Following the principle above, the numbers of channels for MSNAS-R18, MSNAS-R34, and MSNAS-R50 are set to 180, 160, 144 respectively.

## 4.2 RESULTS

Table 1 shows the main results of MSNAS comparing with FPN counterparts. As we can see in the table, best searched architectures of MSNAS achieve mAP at 37.2%, 39.2%, 39.5% at the comparable computation with ResNet18-FPN(34.6%), ResNet34-FPN(37.7%) and ResNet50-FPN(38.3%), with a remarkable improvement of 2.6%, 1.5%, 1.2% mAP gain. Moreover, the best samples in all the experiments outperform the manually-designed baseline networks on average with a relatively small variance. In particular, the performance of the best sample of MSNAS-R18 is comparable with ResNet34-FPN, while the computation of the former one is 20% less than that of the latter one. Also, the maximum of mAP of top samples in the search space of MSNAS-R34 is superior over ResNet50-FPN.

**Computation cost** The super-nets are trained with 4x-schedule for around 30 hours on 16 GPUs. And the evolutionary search stage costs around 3 hours per iteration and about 60hours in total. Then around 90 hours are spent to search the optimal architectures. It could be further improved if better schedules and strategies for training detectors from scratch are proposed.

## 4.3 ABLATION STUDY

**Effectiveness of feature utilization and feature generation search** To verify the effectiveness of feature utilization, we conduct experiments to compare fixed predefined feature selection and our proposed search-based feature selection. For fixed predefined FPN-style feature utilization, all sub-networks in the search space extract feature maps following the same strategies as FPN. Results are shown in Table 2. Feature utilization search in MSNAS shows large improvement compared with the pre-defined feature selection way in the original FPN. By comparing performances of ResNet18-FPN and FPN-style searched architectures, a +0.7%mAP performance gain is obtained by searching stride for encoder network. We find that the correlation between the one-shot performances and stand-alone samples is weaker for the experiment in FPN-style feature utilization, according to Kendall's tau listed in Table 2. We infer that the reason is the discontinuity among paths inside the super-net intensifies.

Table 2: Ablation experiments of pre-defined FPN-style and searched feature utilization.

| Encoder | Feature Utilization | Mean mAP | Var mAP | Max mAP | Kendall's tau |
|---------|---------------------|----------|---------|---------|---------------|
| ResNet18 | FPN | - | - | 34.6 | - |
| Searched | FPN-style | 34.98 | 0.061 | 35.3 | -0.2247 |
| Searched | Searched | 36.94 | 0.03 | 37.2 | 0.4495 |

Table 3: Ablation experiments of stride range constraints

| Stride Constraints | Mean mAP | Var mAP | Max mAP |
|--------------------|----------|---------|---------|
| yes | 36.94 | 0.03 | 37.2 |
| no | 36.98 | 0.107 | 37.4 |

**Feature utilization search space constraints** Several constraints of selected features' resolution are applied when searching feature utilization, as described in Section 3.3. Table 3 shows experiment results with or without stride range constraints. It can be found that although adding constraints cannot achieve much gain of performance but can reduce the variations of performance. Besides, the evolutionary process converges faster with constraints and the variance of one-shot performances in the population is reduced if constraints are added, as shown in Figure 3.

**Fine-tuning strategy** In Table 4, Kendall's taus are computed by the one-shot performances from super-net after fine-tuning and the stand-alone performances of ten random samples with the same computation as the target FLOPs. It can be easily observed that the ranks have a better performance after fine-tuning several iterations in both MSNAS-R18 and MSNAS-R34. MSNAS-R18 with fine-tuning achieves +0.3 mAP at average performance and MSNAS-R34 obtains a +0.6 mAP gain at both the maximal and average performance of top-5 samples.

**Random children search strategy** As noted in Section 3.4, several random children are added to the population for better exploration. According to Table 5, improvement in MSNAS-R50 can be observed by including random children. At the same time, the average one-shot performances of top-5 samples increase by more than 0.1 mAP, which is relatively remarkable among low values of one-shot performances.

## 4.4 COMPARING WITH OTHER METHODS

In Table 6, comparison with other algorithms is conducted. An outstanding performance of 40.7%mAP is achieved by MSNAS-R50. The network of MSNAS-R50-Mask-RCNN and MSNAS-R50* are trained for 6x-long in order to get comparable performance with that of 2x-schedule with pre-trained models. R50-FPN-Faster R-CNN(heavy head) is trained for 6x-long with SyncBN and the bounding-box head at the second stage follows the 4conv-1fc format as noted in He et al. (2019). In order to perform a more fair comparison, we reproduced NAS-FPN. It is trained with

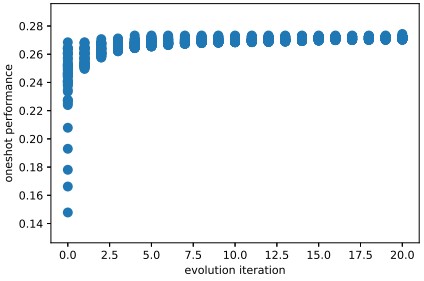
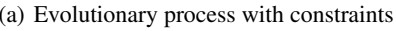
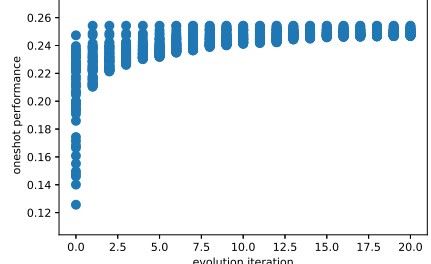

(a) Evolutionary process with constraints  (b) Evolutionary process without constraints

Figure 3: Comparison of the evolutionary process in the ablation of constraints. It's easy to observe that the evolutionary process converges faster with constraints.

Table 4: Ablation experiments of fine-tuning strategy

| Network | Fine-tuning | Kendall's tau | Mean mAP | Var mAP | Max mAP |
|---------|-------------|---------------|----------|---------|---------|
| MSNAS-R18 | yes | 0.4495 | 37.1 | 0.008 | 37.2 |
| MSNAS-R18 | no | -0.0899 | 36.84 | 0.1024 | 37.2 |
| MSNAS-R34 | yes | 0.5683 | 38.86 | 0.0784 | 39.2 |
| MSNAS-R34 | no | 0.2501 | 38.28 | 0.0936 | 38.6 |

Table 5: Ablation experiments of random children strategy

| Network | Random Children | One-shot mAP | Mean mAP | Var mAP | Max mAP |
|---------|-----------------|--------------|----------|---------|---------|
| MSNAS-R18 | yes | 27.12 | 37.1 | 0.008 | 37.2 |
| MSNAS-R18 | no | 27.31 | 36.94 | 0.03 | 37.2 |
| MSNAS-R50 | yes | 25.99 | 39.3 | 0.028 | 39.5 |
| MSNAS-R50 | no | 25.85 | 38.84 | 0.0064 | 38.9 |

weights pre-trained on ImageNet for 2x-schedule. We can see that MSNAS-R50 has an advantage over R50-NAS-FPN(7@256) at a comparable computation.

## 5 CONCLUSION

By analyzing the commonly-used detection framework FPN, we find it critical to generate better multi-scale features and select proper features for detection heads. Considering the fact that multi-scale feature production plays an important role in object detection, we propose a one-shot-based method to efficiently search a complete multi-scale feature generation strategy in the generalized detection architecture. Instead of only modifying network architecture for feature generation, we jointly optimize feature generation and feature utilization. The searched architectures achieve an outstanding performance compared with the state-of-the-art algorithms. More exploration and improvement could be carried out by further works.

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

Table 6: Comparison with state-of-the-art algorithms. For details, please refer to the appendix.

| Model | Encoder | Input Size | FLOPs | mAP |
|-------|---------|------------|-------|-----|
| Faster R-CNN | ResNet50-FPN | 800×1333 | 207.6G | 36.8 |
| Faster R-CNN | ResNet101-FPN | 800×1333 | 290.4G | 39.1 |
| Faster R-CNN(heavy head) | ResNet50-FPN | 800×1333 | 322.8G | 39.1 |
| DetNAS (Chen et al., 2019) | - | 800×1333 | - | 42.0 |
| NAS-FPN R50(7@256) | ResNet50-NAS-FPN | 800×800 | 562.5G | 41.2 |
| NAS-FPN R50(7@256) | ResNet50-NAS-FPN | 600×1000 | 317.9G | 39.7 |
| CR-NAS (Liang et al., 2020) | CR-ResNet101 | 800×1333 | - | 40.2 |
| MSNAS | MSNAS-R50* | 800×1333 | 308.4G | 40.7 |
| MSNAS-Mask-RCNN | MSNAS-R50 | 800×1333 | 376.8G | 41.7(box) 37.5(mask) |

*Advances in Neural Information Processing Systems 32: Annual Conference on Neural Information Processing Systems 2019, NeurIPS 2019, 8-14 December 2019, Vancouver, BC, Canada*, pp. 6638–6648, 2019. URL `http://papers.nips.cc/paper/8890-detnas-backbone-search-for-object-detection`.

Xianzhi Du, Tsung-Yi Lin, Pengchong Jin, Golnaz Ghiasi, Mingxing Tan, Yin Cui, Quoc V. Le, and Xiaodan Song. Spinenet: Learning scale-permuted backbone for recognition and localization. *CoRR*, abs/1912.05027, 2019. URL `http://arxiv.org/abs/1912.05027`.

Golnaz Ghiasi, Tsung-Yi Lin, and Quoc V. Le. NAS-FPN: learning scalable feature pyramid architecture for object detection. In *CVPR*, pp. 7036–7045. Computer Vision Foundation / IEEE, 2019.

Zichao Guo, Xiangyu Zhang, Haoyuan Mu, Wen Heng, Zechun Liu, Yichen Wei, and Jian Sun. Single path one-shot neural architecture search with uniform sampling. *CoRR*, abs/1904.00420, 2019.

Kaiming He, Georgia Gkioxari, Piotr Dollár, and Ross B. Girshick. Mask R-CNN. In *ICCV*, pp. 2980–2988. IEEE Computer Society, 2017.

Kaiming He, Ross B. Girshick, and Piotr Dollár. Rethinking imagenet pre-training. In *ICCV*, pp. 4917–4926. IEEE, 2019.

Feng Liang, Chen Lin, Ronghao Guo, Ming Sun, Wei Wu, Junjie Yan, and Wanli Ouyang. Computation reallocation for object detection. In *ICLR*. OpenReview.net, 2020.

Tsung-Yi Lin, Michael Maire, Serge J. Belongie, James Hays, Pietro Perona, Deva Ramanan, Piotr Dollár, and C. Lawrence Zitnick. Microsoft COCO: common objects in context. In *ECCV (5)*, volume 8693 of *Lecture Notes in Computer Science*, pp. 740–755. Springer, 2014.

Tsung-Yi Lin, Piotr Dollár, Ross B. Girshick, Kaiming He, Bharath Hariharan, and Serge J. Belongie. Feature pyramid networks for object detection. In *CVPR*, pp. 936–944. IEEE Computer Society, 2017.

Tsung-Yi Lin, Priya Goyal, Ross B. Girshick, Kaiming He, and Piotr Dollár. Focal loss for dense object detection. *IEEE Trans. Pattern Anal. Mach. Intell.*, 42(2):318–327, 2020.

Chenxi Liu, Liang-Chieh Chen, Florian Schroff, Hartwig Adam, Wei Hua, Alan L. Yuille, and Fei-Fei Li. Auto-deeplab: Hierarchical neural architecture search for semantic image segmentation. In *IEEE Conference on Computer Vision and Pattern Recognition, CVPR 2019, Long Beach, CA, USA, June 16-20, 2019*, pp. 82–92. Computer Vision Foundation / IEEE, 2019a. doi: 10.1109/CVPR.2019.00017. URL `http://openaccess.thecvf.com/content_CVPR_2019/html/Liu_Auto-DeepLab_Hierarchical_Neural_Architecture_Search_for_Semantic_Image_Segmentation_CVPR_2019_paper.html`.

Hanxiao Liu, Karen Simonyan, and Yiming Yang. DARTS: differentiable architecture search. In *7th International Conference on Learning Representations, ICLR 2019, New Orleans, LA, USA, May 6-9, 2019*. OpenReview.net, 2019b. URL `https://openreview.net/forum?id=S1eYHoC5FX`.

Hanxiao Liu, Andrew Brock, Karen Simonyan, and Quoc V. Le. Evolving normalization-activation layers. *CoRR*, abs/2004.02967, 2020.

Wei Liu, Dragomir Anguelov, Dumitru Erhan, Christian Szegedy, Scott E. Reed, Cheng-Yang Fu, and Alexander C. Berg. SSD: single shot multibox detector. In *ECCV (1)*, volume 9905 of *Lecture Notes in Computer Science*, pp. 21–37. Springer, 2016.

Xin Lu, Buyu Li, Yuxin Yue, Quanquan Li, and Junjie Yan. Grid R-CNN. In *CVPR*, pp. 7363–7372. Computer Vision Foundation / IEEE, 2019.

Bharat Singh and Larry S. Davis. An analysis of scale invariance in object detection SNIP. In *2018 IEEE Conference on Computer Vision and Pattern Recognition, CVPR 2018, Salt Lake City, UT, USA, June 18-22, 2018*, pp. 3578–3587. IEEE Computer Society, 2018. doi: 10.1109/CVPR. 2018.00377. URL http://openaccess.thecvf.com/content_cvpr_2018/html/ Singh_An_Analysis_of_CVPR_2018_paper.html.

Mingxing Tan, Ruoming Pang, and Quoc V. Le. Efficientdet: Scalable and efficient object detection. *CoRR*, abs/1911.09070, 2019.

Ning Wang, Yang Gao, Hao Chen, Peng Wang, Zhi Tian, and Chunhua Shen. NAS-FCOS: fast neural architecture search for object detection. *CoRR*, abs/1906.04423, 2019.

Hang Xu, Lewei Yao, Zhenguo Li, Xiaodan Liang, and Wei Zhang. Auto-fpn: Automatic network architecture adaptation for object detection beyond classification. In *ICCV*, pp. 6648–6657. IEEE, 2019.

Lewei Yao, Hang Xu, Wei Zhang, Xiaodan Liang, and Zhenguo Li. SM-NAS: structural-to-modular neural architecture search for object detection. *CoRR*, abs/1911.09929, 2019.

Yuge Zhang, Zejun Lin, Junyang Jiang, Quanlu Zhang, Yujing Wang, Hui Xue, Chen Zhang, and Yaming Yang. Deeper insights into weight sharing in neural architecture search. *CoRR*, abs/2001.01431, 2020.

Barret Zoph and Quoc V. Le. Neural architecture search with reinforcement learning. In *ICLR*. OpenReview.net, 2017.

Barret Zoph, Vijay Vasudevan, Jonathon Shlens, and Quoc V. Le. Learning transferable architectures for scalable image recognition. In *CVPR*, pp. 8697–8710. IEEE Computer Society, 2018.

## A Appendix

### A.1 Selected Stride of Encoder Network and Feature Stride of Detection Heads

We present the architecture and performance of top-5 samples of MSNAS-R18, MSNAS-R34, MSNAS-R50 in Table 7, Table 8 and Table 9. The stride of blocks in the encoder network could be 0.5(upsample), 1, and 2. The searched architectures do not follow the downsample-upsample style of FPN. Upsampling blocks are more possible to be observed around the middle parts of the networks.

Also, different features of detection heads are selected using MSNAS. Features with absolute stride of 4 and 64 are not utilized for detection heads in most of the best samples.

Table 7: Architecture of top5 samples of MSNAS-R18

| Stride of Blocks in the Encoder | Selected Feature Stride | Performance |
|---|---|---|
| 1, 2, 2, 1, 2, 1, 0.5, 1, 1, 0.5, 1, 1, 0.5 | 8, 8, 16, 16, 32 | 36.7 |
| 1, 2, 2, 2, 0.5, 2, 0.5, 1, 0.5, 1, 0.5, 2, 2 | 8, 8, 16, 16, 32 | 36.8 |
| 1, 2, 2, 2, 1, 1, 0.5, 1, 0.5, 0.5, 2, 1, 2 | 8, 8, 16, 16, 16 | 37.0 |
| 1, 2, 2, 2, 0.5, 2, 0.5, 1, 1, 0.5, 1, 0.5, 2 | 8, 8, 16, 16, 32 | 37.0 |
| 1, 2, 0.5, 2, 1, 2, 1, 2, 1, 0.5, 1, 0.5, 2 | 8, 8, 16, 16, 32 | 37.2 |

Table 8: Architecture of top5 samples of MSNAS-R34

| Stride of Blocks in the Encoder | Selected Feature Stride | Performance |
|---|---|---|
| 1, 2, 1, 2, 1, 1, 2, 1, 0.5, 0.5, 1, 1, 2, 1, 0.5, 1, 0.5, 2, 2, 2, 1 | 4, 8, 16, 32, 32 | 39.0 |
| 1, 2, 1, 2, 1, 2, 1, 1, 0.5, 0.5, 1, 1, 2, 1, 0.5, 1, 0.5, 2, 2, 2, 2 | 4, 8, 16, 32, 64 | 38.7 |
| 1, 2, 1, 2, 1, 2, 1, 1, 0.5, 0.5, 1, 1, 2, 1, 0.5, 1, 0.5, 2, 2, 2, 2 | 4, 8, 16, 16, 64 | 38.4 |
| 1, 2, 1, 2, 1, 1, 2, 1, 2, 1, 0.5, 0.5, 1, 1, 2, 0.5, 0.5, 0.5, 1, 2, 2 | 4, 8, 16, 32, 32 | 39.2 |
| 1, 2, 1, 2, 1, 2, 1, 1, 0.5, 0.5, 1, 1, 2, 1, 0.5, 1, 0.5, 2, 2, 2, 0.5 | 4, 8, 16, 32, 32 | 39.0 |

Table 9: Architecture of top5 samples of MSNAS-R50

| Stride of Blocks in the Encoder | Selected Feature Stride | Performance |
|---|---|---|
| 1, 2, 1, 2, 1, 1, 2, 1, 1, 1, 1, 2, 0.5, 0.5, 1, 0.5, 0.5, 2, 2, 1, 2 | 8, 8, 16, 32, 32 | 39.4 |
| 1, 2, 1, 2, 1, 1, 2, 1, 1, 1, 1, 2, 0.5, 0.5, 1, 0.5, 0.5, 2, 1, 2, 2 | 8, 16, 32, 32, 32 | 39.1 |
| 1, 2, 1, 2, 1, 1, 2, 1, 1, 1, 1, 2, 0.5, 0.5, 1, 0.5, 0.5, 2, 1, 2, 2 | 8, 8, 32, 32, 64 | 39.5 |
| 1, 2, 1, 2, 1, 1, 2, 1, 1, 1, 1, 2, 0.5, 0.5, 1, 0.5, 0.5, 2, 2, 1, 2 | 8, 8, 16, 32, 64 | 39.4 |
| 1, 2, 1, 2, 1, 1, 1, 2, 1, 1, 0.5, 0.5, 1, 0.5, 2, 2, 2, 2, 0.5, 1, 1 | 8, 8, 32, 32, 32 | 39.1 |

