# OpenReview forum: "Multi-scale Network Architecture Search for Object Detection"
_ICLR.cc/2021/Conference — Reject_

### Official Review · AnonReviewer4 · 2020-10-22
**Major concern is about the experimental results**

**Rating:** 5
**Confidence:** 4

**Review:**

This paper proposes a one-shot based multi-scale features ﻿generation and utilization framework for object detection. This method searches the ﻿network stride for features generation and detection heads location for features utilization.

 Pros:
Stride and detection heads location are important factors, and automated learning of them is desirable.
This method improves the performance based on several baselines.
Searching the detection heads location is a new idea, to my knowledge.

Cons:
However, there are some concerns about this paper.
1.     This paper seems like an improved version of the SpineNet. Searching for ﻿strides instead of permutation and utilizing a one-shot method instead of reinforcement learning is not very novel.
2.     If we treat this paper as the improved version of SpineNet, the improvement of the performance is strange. SpineNet improves the performance from 37 to 42.7 for ResNet50. Why jointly searching the backbone features generation and FPN features utilization only improves 1.2 mAP in this paper?
3.     The comparison with other state-of-the-art methods is unfair. To my knowledge, the results for DetNas (42.0) and CR-NAS (40.2) are based on 1x schedule while this method is trained from scratch for 6x schedule.
4. It would be good to evaluate on another dataset.

---

> ### Author Response · Authors · 2020-11-21
> **Response to Reviewer4**
>
> **Q1: Comparison with SpineNet.**
>
> The differences have been discussed in detail in the related work section of the paper. Here we highlight several major distinctions between MSNAS and SpineNet. First, the search space of MSNAS is designed that each operator in the network can downsample or upsample instead of permutation in SpineNet, which builds a much larger search space than SpineNet. Second, the complete multi-scale feature production is considered in our work, while SpineNet only focuses on the architecture of the encoder network. Lastly, our method is based on the one-shot search method instead of reinforcement learning in SpineNet. Our method is much more efficient and requires much less computation cost than SpineNet.
>
> To our knowledge, no previous work has attempted to formulate and design NAS for multi-scale networks. Besides, it does matter to employ one-shot methods to conduct such a search. First, such an efficient and practical NAS algorithm is important for the application. Second, there exist lots of problems to utilize the original SPOS method. We introduce the fine-tuning strategy for evaluation and the random children strategy for search, which are all essential as shown in the ablation experiments.
>
>
>
> **Q2: About the improvement compared with SpineNet.**
>
> SpineNet has a very different schedule from our method. Most of the results of SpineNet are reported on the 20x or 30x schedule with other tricks. Results in our paper are conducted on the common schedule, 6x from scratch. Given the fact that its architectures are trained about 5 times of ours, we don’t think it’s fair to compare the performances and improvement.
>
>
>
> **Q3: About the schedule of results.**
>
> DetNas and CR-NAS use ImageNet pre-training and our method is training from scratch. Empirically, detectors trained for 6x-long from scratch have similar performance as those on the schedule of 2x long with pre-trained weights. For more details, please refer to [1]. For CRNAS and DetNAS, we just utilize the performance reported in the original papers and they both employ ImageNet-pretrained weights. Gaps between 1x and 2x schedule is relatively minor. Therefore the comparison is fair.
>
> [1] Rethinking ImageNet Pre-training

---

### Official Review · AnonReviewer1 · 2020-10-28
**Good results with limited novelty**

**Rating:** 4
**Confidence:** 4

**Review:**

This paper argues searching for both encoder and anchor assignment (feature utilization) in a unified NAS framework would lead to better detector performance. Experiments have been conducted on COCO dataset to show proposed method outperform baseline FPN by a significant margin, and both searching for better encoder and feature utilization benefit object detection task.

Pros:

- dealing with objects with different scale is a fundamental problem in detection, and research in this direction would benefit vision community.

- Most NAS work on detection focus on the backbone feature extractor (encoder). This paper brings new perspective for NAS on object detection task.

- The experimental results back authors' claim that searching for feature utilization brings more performance gains than searching for encoder alone.

Cons:

- The proposed method for searching a path throw super-net across multiple stride is not new. E.g.  "Auto-DeepLab:
Hierarchical Neural Architecture Search for Semantic Image Segmentation" has adopted similar techniques.

- While searching for feature utilization looks novel, from Figure 2 b) it is effectively searching for how to select and fuse features with different resolution to each detector head. Assigning objects with different scales has been explored by works such as "Scale-Aware Trident Networks for Object Detection" / "Feature Selective Anchor-Free Module for Single-Shot Object Detection" in a non-NAS setting, which might be interesting to compare the proposed method with. Also the proposed approach is essentially similar to works that search for FPN connections, e.g. "NAS-FCOS: Fast Neural Architecture Search for Object Detection".

- [minor] anchor-free object detection has growing more popularity in detection tasks, and thus might limit the importance of searching for feature utilization.

- This paper is a bit challenging to follow and would benefit from careful proofreading. See comments below for more details.

Other comments:

- Abstract: "we show that more possible architectures of encoder network and different strategies of feature utilization". by "more possible architectures" does it mean "larger search space"?

- Intro, paragraph 1 "the key to solving" -> "the key to solve"

- Intro, paragraph 2 after introducing FPN / SSD and how they deal with multiple scales, "The basic idea to deal with the multi-scale detection problem can be summarized as below" and re-evaluate FPN architecture. this feels a bit redundant and doesn't read smoothly.

- Intro, paragraph 3 "Also, the predefined rule of feature utilization is very empirical and other alternatives may lead to better performance". Agree with the statement and there has been quite a few work that tries to better align object scales with feature maps (like the papers mentioned above). Please cite and compare.

- Related work. The method proposed in this paper is based on one-shot search and should be simpler than SpineNet. It would be great to have SpineNet in comparison (specially when SpineNet seem to show better performance)?

- Sec 3.1: This section reiterate the motivation of this paper, which have been stated in Intro section and feels a bit redundant.

- Sec 3.2, paragraph 2. "Considering operators with different strides within a mixed-block and the variation of sizes output from different operators in one mixed-block" this is very confusing. Does it indicate we have additional upscaling / downscaling operations within a mixed-block? How this related to searching through a path in a super-net?

- Sec 3.2, Eq 3. Should it be $stride_{j+1} \ne 1$?

- Sec 3.3 last paragraph "descrbied" -> "described"

- Sec 3.3. $s_i$ here indicates the selected feature size w.r.t. to image size (actually should be the inverse of it? otherwise it would be 1/4, ..., 1/64). It is also unclear how this setup would facilitate using multiple feature maps for one head.

- Sec 3.4 "It’s difficult to combine features with different resolutions by element-wise addition, so one-shot based search strategies show great compatibility with our search space." It is not straightforward to understand the connection between combining feature maps at different resolution, proposed scheme and one-shot search strategy. Please elabrate.

- Sec 3.4 " although the primitive weights in the super-net don’t perform well in terms of ranking random samples". This is very confusing. Could authors clarify?

- Sec 3.4 "statics of batch norm..." it should be "statistics"?

- Sec 4: first paragraph. the experiment setup (dataset, splits) is mentioned in appendix but would be better to place it here.

- Sec 4.1 first paragraph "m=in" what does this mean here?

- Table 1: the table is a bit confusing. It compares ResNet-FPN with MSNAS-Resnet. It might be more clear to indicate the backbone network on each row and set the first column to be Resnet-FPN and others for MSNAS-Resnet

- Table 1: Both baseline and proposed method share the same FLOPs. Is this the case? please clarify.

---

> ### Author Response · Authors · 2020-11-23
> **Response to Reviewer1 Part 2/2**
>
> Other Comments:
>
> Thanks again for the suggestions on writing and we will fix the typo and grammar issues in the updated version.
>
> 1) “more possible architectures” means more new designs of different detection networks obtained by MSNAS. For example, the architecture of FPN follows the decreasing-increasing style, but we find more powerful architectures with different internal stride changes.  Design of networks manually requires substantial efforts and our method provides an efficient way.
>
> 2) We have stated a comparison about methods with SpineNet in details in the related work section. For performance comparison, SpineNet reports its results on 20x, a very different schedule compared with ours and most of the object detection methods. And it is hard for us to follow or even reproduce the performances of the searched architectures. Therefore, it’s hard for us to conduct a fair comparison. That’s why we don’t include the results in Table 6.
>
> 3) For clear description, we formulate the super-net composed of mixed-blocks and each mixed-block consists of blocks with different stride values. Our method is path-wise, so the super-net is treated as a sequence of N mixed-blocks and each architecture is a sequence of N blocks. In the implementation, there isn’t an actual object of a mixed-block. The super-net is comprised of 3*N blocks. In each iteration during training, one of the three candidate blocks within one mixed-block is selected to form one path, whose parameters will be updated.
>
> 4) Thanks for pointing out the typo here. It should be $stride_{k+1}$. We’ll correct it in the updated version.
>
> 5) $s_{i}$ means the quotient of the input image’s size divided by the box’s size. This can be inferred from Eq.4 and the example in the second paragraph in Sec3.3.
>
> 6) Traditionally, NAS methods follow three types, RL-based ones, differentiable ones and one-shot based ones. RL-based methods always require large quantities of computation resources but we aim to propose a more efficient one. So we don’t employ RL to implement our algorithm. In differentiable methods, candidate operators are weighted summed so that the best operator could be chosen according to the weights of arch parameters. For more details about differentiable NAS, please refer to [1], [2], and other works. We mention the difficulty of combining feature maps with different resolutions to explain why we don’t use the differentiable methods. For one-shot NAS, there is no explicit connection between operators within the same mixed-block, so by following the one-shot style, we avoid the summation of feature maps with different resolutions. For more information please refer to [3] and other related works.
>
> 7) One-shot NAS utilizes the strategy of weight-sharing. To be more specific, one-shot NAS methods employ weights from the super-net to evaluate the architectures, which could be treated as one type of proxy. The evaluated performances of the architectures are used for selection in EA. Therefore, the ranking of samples using weights from the super-net is important for selecting well-performed architectures. Unpleasantly, the super-nets of MSNAS always have poor ability to rank random samples so we utilize the fine-tuning strategy to obtain a better ranking of architectures.
>
> 8) The details of FLOPs are described in detail in the appendix. In Table1, MSNAS and its FPN counterparts share the same FLOPs of the encoder networks and RPN. The FLOPs of the entire network are a little different and MSNAS has marginally smaller FLOPs than its counterpart, which can be observed from Table 6.
>
>
>  [1] DARTS: Differentiable Architecture Search
>
>  [2] ProxylessNAS: Direct Neural Architecture Search on Target Task and Hardware
>
>  [3] Single Path One-Shot Neural Architecture Search with Uniform Sampling

---

> ### Author Response · Authors · 2020-11-23
> **Response to Reviewer1 Part 1/2**
>
> **Q1: Comparison with Auto-Deeplab.**
>
> There are several differences between Auto-Deeplab and our work. First and foremost, we aim to design multi-scale networks, while Auto-Deeplab has a single output feature map which may undermine the influence of changing internal stride values. Second, Auto-Deeplab utilizes differentiable search methods, but our work is one-shot based. Both of the search methods are widely used in NAS but rather different for algorithm design and implementation. Third, we focus on improving multi-scale detection and Auto-Deeplab aims to achieve better semantic segmentation.
>
> What’s more, we model the production of multi-scale features as two steps, but feature utilization problem has not been considered in Auto-Deeplab. As shown in our experiments, feature utilization is very important to multi-scale networks and contributes to a remarkable improvement in our work.
>
> **Q2: Comparison with other object assigning methods.**
>
> Although these methods and our work all attempt to deal with the multi-scale detection problem, they have different motivations and try to solve the problem from very different perspectives.
>
> FSAF proposes online feature selection to improving the training of the anchor-free branch. In our work, once the searching process ends, the feature utilization strategy is determined as one part of the network, and the anchors and proposals are assigned according to the searched strategy during both training and inference time.
>
> TridentNet is based on the observation that the receptive field influences the detection performance and utilizes three separated branches with different dilation rates. However, our work pays more attention to feature quality and matching object groups and feature maps.
>
> NAS-FCOS searches for the operators and connections of the FPN and the FCOS head. For operators, we don’t involve operator diversity in our work. For connections, FCOS conducts a search on feature fusion of features generated by the backbone and adopts a fixed feature utilization strategy. The basic idea of the connection search in FCOS has some common parts with NAS-FPN. And  NAS-FCOS works based on FCOS and we conduct our experiments based on FPN. So we NAS-FCOS is not included in our paper and we’ll add it to Table 6.
>
> Basically, these methods are mainly improvements of object detection framework, which have large differences with our proposed NAS method.
>
> **Q3: About anchor-free methods.**
>
> Feature utilization serves to assign objects in the context of improving FPN, but it’s not limited to assigning anchors. Anchor-free frameworks are also built on multi-scale networks so that MSNAS can also be applied to anchor-free methods. By searching for feature utilization, we aim to find the strategies of relating objects with feature maps. For example, FCOS employs the feature maps of P3-P7 to determining objectiveness with the pre-defined rules. With MSNAS, FCOS has the opportunity to obtain a better set of feature maps than those from P3-P7. Therefore, anchor-free methods won’t undermine the importance of our work, since MSNAS is built on the basic structure of multi-scale detection.
>
> **Q4: About writing.**  Thanks a lot for pointing out the typos and errors and we will modify them in the updated version.

---

### Official Review · AnonReviewer2 · 2020-10-28
**Ok but not good enough - rejection**

**Rating:** 4
**Confidence:** 3

**Review:**

#### Summary

This submission works on the task of architecture search for object detection. The authors focus on two components: how to produce multi-scale features and how to use multi-scale features. The authors formalized a simple search space, and applied an evolution-based search algorithm. Experiments show the proposed searching algorithm is able to outperform the FPN baselines with various (small) backbones.

#### Strengths

- The task of multi-scale feature modeling in object detection is indeed important.

- The proposed searching method is straightforward and easy to understand, and the performance compared to the baseline is OK.

- Figure 1 is good. It clearly summarized the contribution.

#### Weaknesses

- A clear limitation is the authors did not compare to other NAS-based FPN as discussed in the related work section. Only NAS-FPN is listed in Table. 6, and it outperforms the proposed method (despite heavier computation). In my opinion, the most important comparison is to BiFPN, which also has similar FLOPS as FPN and performs 2-4 mAP higher according to EfficientDet paper.

- The second major complaint is the scalability. The authors only show backbones up to Res 50, but not stronger ones. For example, ResNeXt101-DCN is used for reporting the state-of-the-art number in many detection paper. Ideally, the author should show their method works on this large backbone to really push the state-of-the-art. If the computation is a problem (The author already used 16 GPUs, which is more than the "standard budget" in object detection frameworks (8 GPUs)), then it's a limitation to the method.

- The authors claimed changing the backbone encoder as an advantage of the proposed method. I would argue that this also slightly makes the proposed method less practical: using a fixed backbone can better utilize the pretrained weights, and can better use the community progress in designing backbone networks.

- While the overall idea is straightforward, it also makes the paper lack excitement. While I am not working on NAS, the paper makes me feel it just applies a NAS recipe to multi-scale feature production (I can see designing the search space is a contribution). Please correct me if I am wrong.

- (Minor) The authors only evaluate on two-stage Faster/ Mask RCNN, where the object level assign is less critical (e.g., PointRend assigns all objects to stride 4 level). I would suggest the author also try one-stage detectors, where the effectiveness of FPN is more pronounced (e.g., FCOS shows subtle differences in FPN level assign yield 1-2 mAP difference). Note that this is not a requirement for rebuttal.

#### Summary

This paper applies a reasonable method to an important problem. The results are OK (decently outperforming the baseline) but not exciting (lack comparison to the state-of-the-art). Also, the computation requirement may limit further applying the proposed method on large backbones. In its current status, I suggest a rejection. In the rebuttal, if the authors clearly show the searched structure outperforms BiFPN and works on large backbones, I will improve my rating.


#### After rebuttal

Q1: Thanks for running the additional experiments. Unfortunately, the results are not strong enough to convince me to use the searched architecture instead of BiFPN. It will also be interesting to see how BiFPN works under similar FLOPs with the searched model (e.g., the structure of EfficientD4 or D3).

Q2: Please do run the ResNeXt101-DCN experiments for the revision or next submission. These are critical to make people use the proposed method.

Q3: True. However training from scratch requires 6x longer training time according to [1], and is considered as a drawback.

Q4: Thank you for the clarification, this makes the contribution clearer. However my concerns on changing the backbone remains (Q3).

Q5: Thank you for considering. I agree the ranks in the leaderboard is a main factor for design choice. This also highlight the importance of Q2.

---

> ### Author Response · Authors · 2020-11-21
> **Response to Reviewer2**
>
> **Q1: Comparison with BiFPN.**
>
> It’s difficult to conduct a fair comparison with BiFPN. BiFPN is not a NAS-based method and the backbone of BiFPN is based on EfficientDet, which employs a totally different search space compared to our method. Also, the configurations of training are quite different from ours. In EfficientDet, most models are trained for 300 epochs with complex augmentations.  We supplement the experiments of ResNet50-BiFPN, as shown below. We adopt the architecture of BiFPN in EfficientD5 and train the network with 800x1333 input using SyncBN for 2x long with pre-trained weights.  Compared with BiFPN-R50, MSNAS-R50 has a lower computation with comparable performance.
>
> |  Method              |       AP            |  FLOPs |
> |----------------------|------------------| ------------------|
> | MSNAS-R50       |             40.7          | 308.4G|
> | BiFPN-R50         |              40.9       | 355.2 G|
>
>
> **Q2: Question about scalability and larger models.**
>
> Experiments on MSNAS-R18, MSNAS-R34, and MSNAS-R50 have already shown the scalability and generalization ability of our method. Larger models do require larger computation cost and memory cost, which is a common limitation for most of the NAS methods on object detection such as NAS-FPN and SpineNet. Compared with these methods, our method has already significantly reduced the computation cost.
>
> As larger models like Res101 and ResNeXt101 require more time to train and a longer schedule to converge, we attempted to supplement a larger experiment but failed to obtain the final results. More experimental results will be available later if possible.
>
> **Q3: Question about changing the backbone encoder.**
>
> Searching for the optimal backbone network for object detection is one of the advantages of our method and we don’t think it will make the proposed method less practical for the following reasons. First, previous works have shown object detection models training from scratch can achieve comparable performance as those fine-tuned from ImageNet pre-trained models [1]. ImageNet pre-training is not necessary for the training of object detection models. Second, a good network architecture on the image classification task is not guaranteed to be a good network architecture for object detection, especially considering the differences between two tasks such as the multi-scale problems in object detection. Therefore it is reasonable to directly search the optimal network on detection tasks.
>
> [1] Rethinking ImageNet Pre-training
>
> **Q4: Question about main contribution and novelty.**
>
> Search for the multi-scale network architecture is a very challenging task. Previous NAS methods mainly focus on the “single output” network. And previous works for NAS on object detection only focus on part of the detection networks, such as only the backbone or only the neck. As far as we know, our proposed method MSNAS is the first work to design NAS for multi-scale networks and optimizes the backbone, neck and detection head together in object detection.
>
> We formulate the multi-scale networks into two jointly optimized parts, feature generation and feature utilization. Previous works focus on designing architectures to generate better features but fail to pay attention to feature utilization. NAS-FPN optimizes the architectures of generating features from outputs of a classification network. SpineNet aims to generate features from a raw image but has a fixed pattern of feature utilization. Moreover, the searching and training cost makes it difficult to follow. Our method is more efficient in searching. Besides, we employ fine-tuning for better weight-sharing strategies and random children during EA to encourage exploration.
>
> **Q5: Extension to other detectors.**
>
> Thanks for the suggestion. We chose two-stage detectors mainly because they are widely used and dominate the top ranks of leaderboards of benchmarks such as COCO. Object assignment is actually also vital to two-stage detectors and the original FPN work is a two-stage object detector. We would like to evaluate our methods on one-stage object detection methods.

---

### Official Review · AnonReviewer3 · 2020-10-29
**Incremental contribution in FPN design element considered in the proposed NAS method**

**Rating:** 3
**Confidence:** 4

**Review:**

This paper introduces a network architecture search (NAS) suitable for a feature pyramid network (FPN) that provides notable detection accuracy for objects at every scale. Based on the decomposition of FPN structure as (multi-scale) feature generation and feature utilization, the proposed NAS offers a new design strategies for both components. For feature generation, NAS super-net is trained to find the optimal selection of whether to reduce, maintain, or extend feature resolution after each module. For feature utilitization, it defines conditions for efficiently selecting the optimal FPN architecture. The proposed MSNAS yields the better accuracy than its backbone FPN and other NAS methods on the COCO object detection benchmark dataset.

I have a concern in technical novelty.
The only novel element considered in the proposed NAS method is to learn the stride value that controls the feature resolution at each module. The most contribution of this paper is very similar to FPN-NAS in that it re-designs FPN by choosing the optimal connections between all the modules. Furthermore, this search method does not retain the main claim of FPN, in which features at every level can be trained to have the same level of semantics.

The method used for Feature utilization is a very simple rule not relying on training to select the final feature layer to be connected to object detection heads. This simple rule can speed up training at the expense of marginal accuracy. This can be seen as part of engineering rather than technical contribution.

I also have a concern in its presentation.
Algorithm 1 is very important for understanding how to implement the proposed method. However, it is very difficult to understand this algorithm due to missing definition of some terms such as CrossoverEncoder, MutationEncoder, CrossoverFeatureStride, MutationFeatureStride etc.

---

> ### Author Response · Authors · 2020-11-21
> **Response to Reviewer3**
>
> **Q1: Comparison with NAS-FPN.**
>
> Our work has major distinctions with NAS-FPN. Detection networks such as FPN can be separated as three components: backbone, neck and head. NAS-FPN only focuses on one part of the detection network, i.e. the neck component while the backbone and detection head are still based on previous human designs. On the contrary, MSNAS is a holistic approach for searching the entire detection network, including backbone, neck and detection head altogether. Our experimental results in Table 2 also prove that the search for the entire network is important for object detection and optimizing only part of them could lead to sub-optimal results.
>
> About the main claim of FPN question, not following the original FPN feature assignment way and instead searching for the optimal feature utilization way is one of the contributions of our method. The assignment of objects with different scales to different feature maps in FPN is quite hand-crafted and rule-based, which could not be the best choice and we aim to improve it by the search-based method. Therefore, the strategies of feature utilization are designed not to follow the FPN-style. And the experimental results have shown that searched feature utilization strategies are indeed better than the original FPN-style.
>
> **Q2: Training of Feature Utilization.**
>
> The method used for feature utilization is indeed based on training and is not a simple rule at all. The strategies of feature utilization and feature generation are jointly optimized as a whole with one-shot NAS optimization method. And the selection of feature utilization is based on weights of the trained super-net, instead of a non-training rule. The feature utilization is nontrivial considering that there are more than 100 candidates of feature utilization given one encoder architecture with the maximal stride is 64 and the minimal one is 4. It’s hard to decide which is better to select if no training methods are involved. As shown in Table 2, only searching for FPN-style architectures would not achieve a great improvement, which shows the effectiveness and importance of the feature utilization method. Thus, searching for feature utilization is far more than a simple rule.
>
> **Q3: Clarification of Algorithm 1.**
>
> We apologize for the misunderstanding notations in Algorithm 1. Algorithm 1 is described in Sec3.4 and we will update the paper with more clear notations. In Algorithm 1, CrossoverEncoder means doing a crossover concerning the stride values in the encoder network. CrossoverFeatureStride means doing a crossover concerning the selected stride values of utilization. MutationEncoder means doing mutation concerning the stride values in the encoder network. MutationFeatureStride means doing mutation concerning the selected stride values of utilization.

---

### Author Response · Authors · 2020-11-23
**Paper Update**

Thanks to all reviewers for their constructive comments and suggestions.

In the updated version, we carefully considered all reviews' suggestions to improve the paper. The updates include the following aspects:

1. Notations and typos. The meaning of CrossoverEncoder in Algorithm1 is explicitly explained in Sec3.4. The typo of the subscript in the equation is corrected. The typos in Sec3.3, Sec3.4, and Sec4.1 have been fixed.
2. Table1 has been re-drawn for better readability.
3. Writing. We have polished content in Sec3.1 and 2nd paragraph in Introduction for reading more smoothly. We add more detailed and formal descriptions of the problem and the processes of feature generation and feature utilization. Besides, we modify the wording of formulating the problem from FPN in the introduction part.

---

### Decision · Program_Chairs · 2021-01-07
**Final Decision**

**Decision:**

Reject

**Comment:**

This submission got 3 rejection and 1 marginally below the threshold. In the original reviews, most of the concerns lie in the limited novelty, the inferior performance to some existing similar works and the limited scalability of the proposed method. Though authors provide some additional experiments, the reviewers still feel the experiments are not convincing and keep their ratings. AC agrees with the reviewers comments on this paper. Though achieving SOTA performance is not necessary for every submission, NAS-alike method is purely pursuing better performance (either higher accuracy or better efficiency). Thus, the performance is also important for evaluating a NAS paper. From the reviewers, the proposed method does not show better performance than some existing works, like BiFPN. This makes the value of the paper is not clear, in particular considering the method novelty is limited. The authors could consider to improve the submission in the experiments to better justify the proposed method, either achieving better performance or higher efficiency than existing works. At its current status, AC cannot make accept recommendation.